# Importance of Osseoperception and Tactile Sensibility during Masticatory Function in Different Prosthetic Rehabilitations: A Review

**DOI:** 10.3390/medicina58010092

**Published:** 2022-01-07

**Authors:** Diego González-Gil, Ibrahim Dib-Zaitum, Javier Flores-Fraile, Joaquín López-Marcos

**Affiliations:** Dental Clinic Faculty of Medicine, University of Salamanca, Paseo Universidad de Coimbra, 37007 Salamanca, Spain; diegoggil@usal.es (D.G.-G.); ibrahimdib@usal.es (I.D.-Z.); jflmarcos@usal.es (J.L.-M.)

**Keywords:** osseoperception, tactile sensibility, interocclusal thickness, interocclusal perception

## Abstract

*Background and Objectives*: Tactile sensibility is an important characteristic for evaluating the masticatory efficiency in different occlusal situations. When a tooth is extracted, relevant proprioceptors from the periodontal ligament get lost; and after the rehabilitation of this abscess by means of oral prosthesis, this sensibility decreases influencing masticatory function. Osseoperception is a sensitive phenomenon associated with dental implants that allows an increased tactile sensibility to those wearing implant prostheses. The purpose of this study was to determine the difference in tactile sensibility values between implant prosthesis, complete dentures, and natural teeth through a review of the available literature. *Materials and Methods*. In order to dissect the information, 24 articles from 2004 to 2021 were analyzed from MEDLINE, PubMed Central, and Web of Science databases. These articles were directly related to measuring tactile sensibility in different situations and demonstrating the influence of osseoperception in an improved masticatory function. *Results*: Tactile sensibility in implant prosthesis is slightly reduced compared with natural dentition but presents improved values with regard to complete dentures. *Conclusions*: Implant prosthesis are more effective during masticatory function than complete dentures, as they present an increased tactile sensibility, very similar to that present in natural dentition. This enhanced sensibility in implants is due to the osseoperception phenomenon.

## 1. Introduction

Tactile sensibility associated with teeth has a great relevance for achieving the proper behavior of the masticatory system. The periodontium has lots of different kinds of mechanoreceptors that are able to detect a low gauge stimulus, such as a small amount of pressure or position modifications. This sensitive phenomenon avoids the execution of great occlusal forces that may damage teeth, allowing a correct oral function. After tooth extraction, the receptors inside the periodontal ligament disappear, as well as the important sensitive information so useful during chewing [1]. For centuries, a lot of different types of prostheses have been designed to replace missing teeth with better or worse results. Complete dentures might be unstable and damage soft tissues, giving the patient unsatisfactory results. Currently, after the emergence, normalization, and protocolization of osseointegrated implants and their rehabilitations, a great alternative to conventional prostheses has been achieved [2] since implant prostheses are related to the osseoperception phenomenon that enables an improvement in functional integration.

Osseoperception consists of the sensation arising from mechanical stimulation of an implant prosthesis transmitted by mechanoreceptors from the masticatory system, along with a modification in central neural processing. Nowadays, the behavior of this sensibility is not well understood, and more clinical studies about this subject are needed. However, it is believed that mechanoreceptors from adjacent tissues are involved, such as masticatory muscles, temporomandibular joint, or gums. It is even thought that residual periodontal mechanoreceptors that remain after dental extraction might be responsible for capturing and transmitting osseoperception [3]. As a result of this mechanical sensibility, a remodeling of the central nervous system occurs in order to adapt itself to a new sensitive and motor function since cortical brain areas that were in charge of taking in some kind of stimulus acquire another sensitive goal by integrating implant-supported prostheses [4].

Anterior studies about osseoperception are aimed at monitoring this mechanical sensibility to see its origin. Specifically, two kinds of investigations are carried out in order to measure the influence of osseoperception: psychophysical studies and neurophysiological studies. Prevailing are the psychophysical studies, as they are able to connect patients’ psychological response with receptors’ sensitive response [4,5].

The theoretical basis of these studies is the implant’s tactile sensibility, which allows, for instance, the achievement of an inhibitory muscle response against traumatic occlusal forces by detecting a really fine stimulus.

Tactile sensibility can also be divided into two types: active tactile sensibility and passive tactile sensibility. Active tactile sensibility gauges patients’ perception when detecting some strange element between teeth during masticatory function, and it is measured in micrometers. On the other hand, passive tactile sensibility consists of the lower force that can be detected when applied to an implant or a natural tooth and is measured in Newtons. Additionally, it is an independent parameter from the patient’s perception. The main difference between these kinds of sensibility is that the passive one only measures individual neuronal receptors that exist on peri-implant areas; while the active one evaluates the whole masticatory function, regarding both receptors near implants and those ones located in more distant areas, such as the temporomandibular joint or masticatory muscles [4].

In this context, the purpose of this study was to determine the differences in tactile sensibility values between conventional complete dentures, implant-supported prostheses, and natural dentition. Regarding implant-supported prostheses, the study analyzed the influence of osseoperception on increased tactile sensibility.

## 2. Materials and Methods

### 2.1. Study Design

A review of tactile sensibility linked to osseoperception phenomenon was carried out, including articles during the period comprising January 2004 to December 2021 covering only articles published in English.

We performed study selection according to the PRISMA (Preferred Reporting Items for Systematic Review and Meta-Analyses) guidelines for reporting systematic reviews. This revision has been registered in PROSPERO (registration code 301737).

The search strategy was conducted using the population, intervention, comparison, and outcome (PICO) framework based on the following question: “Do prostheses on implants have greater osseoperception than conventional prostheses?”

To answer this question, a sample population group of patients undergoing treatment with implant prostheses was selected

### 2.2. Inclusion Criteria

Both research and review articles were included. At first, we added review articles that explained in a clear way the osseoperception phenomenon and its relation between tactile sensibility. Second, we studied research articles consisting of psychophysical and neurophysiological investigations, such as the measurement of active and tactile sensibility in different prosthetic rehabilitations, using interocclusal foils or customized devices.

### 2.3. Exclusion Criteria

Studies concerning the links between osseoperception and implant integration or other parameters not related to tactile sensibility were not included, except for those concerning periodontal neurophysiology; nor did we include those articles in which the full text was unavailable. Finally, 107 articles were rejected from this investigation due to their lack of a relationship with the review objectives or for being duplicates, as shown in the flowchart (Figure 1).

### 2.4. Variables

Among all studies reviewed, the relation between osseoperception and tactile sensibility in natural dentition and some kind of prosthetic rehabilitation was assessed.

### 2.5. Resources

#### Bibliographical Resources

The medical database PubMed–MEDLINE was consulted, as well as Semantic Scholar. Social media ResearchGate was used as a complement in order to obtain some full-text articles under consent from their authors.

Keywords used were “osseoperception”, “tactile sensibility”, “interocclusal perception”, “interocclusal thickness”, and “oral somatosensory function”. A cross-search was also performed using these terms: “osseoperception and tactile sensibility” and “osseoperception and interocclusal thickness”.

## 3. Results

Twenty-five articles were chosen among all the literature reviewed so as to analyze the influence of osseoperception on an improved tactile sensibility and its related values in every prosthetic rehabilitation or natural dentition.

Following, there is a flowchart summarizing article selection (Figure 1) as well as two tables; first one showing all review articles (Table 1) and second one showing all clinical investigations (Table 2). These tables also describe every author, year of publication, type of investigation, study objective, and results.

A first clinical study that demonstrated the presence of nervous fibers around implants in humans was conducted by Corpas et al. [16], showing that it was necessary to extract failed implants from patients’ bone and to analyze them by using a microscope. Research results showed the presence of mielinic and amielinic fibers in Haversian canals of bone tissues surrounding implant threads. In addition to this, there was a study that demonstrated the presence of changes in the cortical brain after rehabilitating dental abscesses with implants [20].

Analyzing tactile sensibility, there was a higher number of investigations of active tactile sensibility than passive tactile sensibility because active tactile sensibility is easier to measure mechanically and better reflects masticatory function clinically.

The main investigations of tactile sensibility thresholds in different prosthetic rehabilitations and natural teeth were those from Enkling [19,21,22,25], Shala [13], Reveredo [17], Kazemi [18], Grieznis [23], or Batista [24]. In conclusion, the thresholds in implant-supported prosthesis were lower than those in complete dentures and very similar to those present in natural dentition, and this fact allowed an increased sensibility in dental implants that facilitated the functional integration of these prostheses [7,8,11].

## 4. Discussion

Nowadays the behavior of osseoperception is not well known, as there are not many studies concerning this subject, especially studies related to active and passive tactile sensibility. Some studies are focused on the mechanisms of osseoperception in the cortical brain [20,28], and others describe how this phenomenon influences somatosensorial perception—for instance, the review from Haggard [29]. At first glance, the review articles are useful for understanding how osseoperception works and the way it is related to tactile sensibility.

Among all of the reviews selected referring to osseoperception, the studies of Mishra and Bhatnagar stood out. Mishra [2] concludes that osseoperception is the phenomenon that causes an improved functional integration of dental implants, which is represented by an increased tactile sensibility. Bhatnaghar’s review [1] also states that there is a clear improvement in function regarding implant-supported prosthesis thanks to osseoperception and tactile sensibility [30].

Other reviews such as those by Abarca, Van Steenberghe, Jacobs, and Kumar [3,4,11,31] are focused on the analysis of osseoperception and its relation to masticatory muscles and its neurophysiological implications. Studies show how an implant prosthesis presents a better physiological and functional integration than complete dentures and that its behavior is more natural, as well as the importance of a correct occlusion [32,33]. However, although their masticatory efficiency is correct, patients with an implant prosthesis suffer more muscle fatigue and lower biting force under extreme effort than those preserving natural dentition [34]. Second, in Feine’s investigation [35], it is explained how the evaluation of masticatory ability in prosthetic rehabilitation is a subjective parameter, suggesting that further studies must rely on patients’ opinions about comfort sensations while wearing a prosthesis rather than its stability.

Tanaka [14] also performed an investigation that concretely measured the influence of osseoperception in oral function by using immediate loading protocols and analyzing the adaptation results after rehabilitation. Results showed a gradual improvement in bite force, not including any improvement in masticatory efficiency or food roughness perception [36]. Regarding studies about tactile sensibility in natural teeth, it is important to analyze Trulsson’s review [9], which is focused on describing periodontal receptors and their relevant proprioceptive function for a proper functioning of a masticatory system that is affected after dental extractions [37,38,39].

Among active tactile sensibility investigations, Enkling’s research stands out [19], which intended to establish a comparison of perception between implants and natural teeth by using fine copper leaf gauges that patients were asked to notice during chewing. Results showed a really low sensibility threshold that was very close to natural dentition. In addition, he stated the great relevance of a correct occlusal setting, as some sensibility values are as low as micrometers in size, which suggests that tiny interferences in occlusion can be perceived by the patient in both implant prosthesis and natural teeth [40].

Another research study concerning active tactile sensibility was conducted by Shala [13], who studied sensibility associated with complete denture carriers. This investigation compared patients who had been wearing prostheses for several years with respect to new carriers, in order to measure the adaptation during the passage of the time. Regarding these results, after wearing a prosthesis for fifteen weeks, the sensibility suffered an improvement, although the sensibility threshold remained three times higher than natural dentition.

Kazemi [18] also investigated active tactile sensibility in implants with respect to natural teeth, concluding that sensibility in implants is slightly reduced compared with natural teeth. In Reveredo’s [17] research, the active tactile sensibility threshold was increased two times with respect to natural dentition.

One of the few available studies about passive tactile sensibility was that of Grieznis’ [23], which compared sensibility between implants and natural teeth. In passive studies, interocclusal leaf gauges are not needed for measuring perception, so direct stimulus was applied on the implants without any kind of patient participation. In this case, the threshold discrepancy was much higher than that regarding active studies, as the implant threshold was also higher. This was due to the lack of participation of remote receptors far away from peri-implant areas, which are not included in passive studies [41,42].

According to the bibliography reviewed, osseoperception is a phenomenon whose existence is proved, yet it is not well or accurately known how it works. In 2005, a consensus statement was published [12] involving several investigators dedicated to the study of osseoperception in which they regarded osseoperception as the mechanical sensibility related to dental implants. This statement suggested that the presence of mechanoreceptors located in muscle, articular, mucous, and periosteal tissues was responsible for this sensibility, along with a neurophysiological change in superior neural centers. Subsequently, there have been more articles studying this phenomenon and its repercussions in correct oral function [43,44].

After the Corpas [16] investigation, the presence of nervous fibers surrounding peri-implant tissues was clear, so that existence of osseoperception became more evident. To be able to make a more accurate statement, more studies about active and passive sensibility are needed, as currently they are very scarce.

There is a consensus about considering that both active and passive tactile sensibility are increased in patients who have been rehabilitated with implant prosthesis compared with those wearing complete dentures, and this is due to osseointegration. Otherwise, differences in sensibility values present large discrepancies between some studies, causing contradictory results in some cases. This happens fundamentally because of the lack of homogeneity in some parameters such as research groups, experimental methods, or statistical approach. For instance, Kazemi [18] describes the wide differences in active sensibility values while studying bibliographical references in which the rank varied from 10 to 100 μm. Generally, sensibility thresholds in implants are higher than those in natural dentition, although they are closer to them than those in complete dentures, as some investigators such as Enkling, Kazemi, or Reveredo assert [17,18,25].

It is interesting to note how a difference between thresholds increases in passive tactile sensibility studies, as in the Grieznis study [23]; this is due to the fact that during this kind of procedure there is an activation of peri-implant proprioceptors, while receptors from other areas such as masticatory muscles or the temporomandibular joint keep quiescent.

There is a general coincidence in every investigation about the subject, and it consists of the really low sensibility threshold in both natural dentition and implant prosthesis and the response to very fine stimuli, as the detection of tiny elements with a few micrometers of thickness becomes possible. If we extrapolate these results to daily clinical performance, it is easy to realize the effects of our rehabilitations in masticatory function. Small occlusal variations can be detected by the patients, and tiny interferences may cause important occlusal alterations. This fact reaffirms the relevance of a correct occlusal settling after performing any prosthetic rehabilitation [23].

## 5. Conclusions

Implant prostheses present an increased tactile sensibility compared with complete dentures, and their values are closer to those in natural dentition, as have been shown in neurophysiological and psychophysical studies. This improved sensibility entails a better masticatory function in patients rehabilitated with implant-supported prostheses. As a result, these rehabilitations are a great alternative to complete dentures when treating edentulism.

Osseoperception is the phenomenon responsible for this upgrade of sensory perception in these kind of rehabilitations, although its operation is not well known yet.

In order to understand osseoperception in detail and to evaluate more precisely the tactile sensibility of implant prostheses, more studies are needed, and these investigations must have outcome criteria that are more homogeneous.

## Figures and Tables

**Figure 1 medicina-58-00092-f001:**
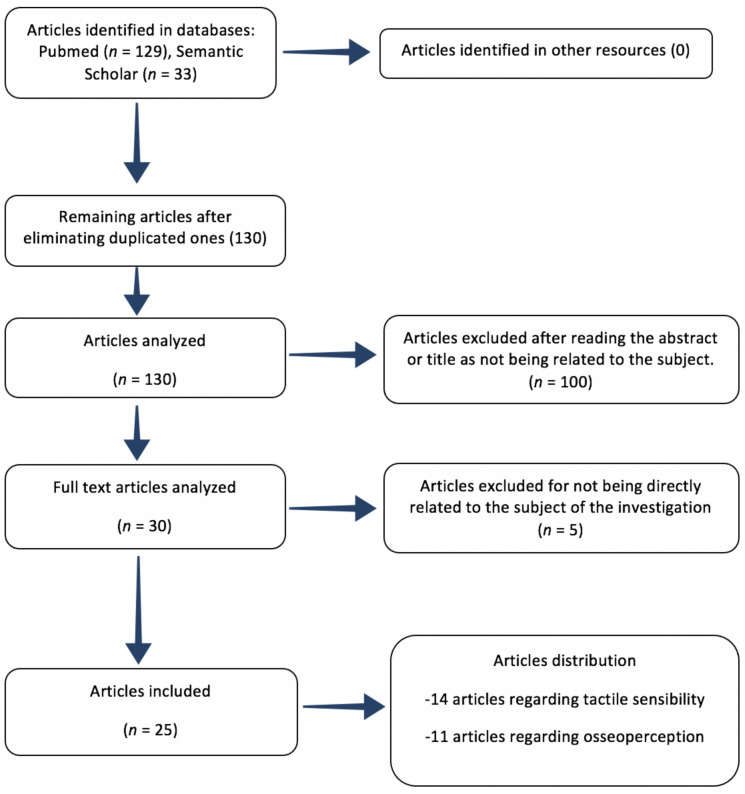
PubMed flowchart summarizing the review process.

**Table 1 medicina-58-00092-t001:** Review article results.

Author and Year	Objectives	Important Study Characteristics	Results
Flanagan [6] 2017	Measuring the importance of the biting force in implant prosthesis so as to obtain a proper rehabilitation.	Medline PubMed literature search with 30 articles discussed.	The biting force was an important parameter during the planification of implant-supported prosthesis, as patients with high load levels may have had failures in the rehabilitation process.
Bhatnagar et al. [1] 2015	Studying the histological, neurophysiological, and psychophysical aspects of osseoperception.	Comprehensive research in PubMed and Google Scholar to retrieve 29 studies from 1985 to 2014.	Dental implants allowed a great satisfaction and clinical function thanks to osseoperception, a phenomenon whose mechanisms are not well understood yet.
Mishra S. et al. [2] 2014	Revising histological, neurophysiological, and psychophysical studies in order to understand how osseoperception and tactile sensibility work.	Review in PubMed database to retrieve 81 articles from 1960 to 2014.	Osseoperception allowed a higher tactile sensibility and a better integration of the implant prosthesis when compared with complete dentures.
Higaki N. et al. [7] 2013	Studying the difference in sensibility between natural teeth and dental implants.	Research in PubMed database of 23 articles and meta-analysis of 6 articles from 1980 to 2012.	Both tactile sensibility and thickness perception presented higher thresholds in implants than in natural teeth.
Kumar et al. [8] 2012	Performing an overview about neurophysiological ability of osseoperception.	Review of 23 studies from 1978 to 2006.	When we rehabilitate edentulism with dental implants, a proper sensibility pathway is created, leading to a better functional adaptation.
Trulsson M. [9] 2006	Describing sensitive and motor function of periodontal receptors.	Review of 47 articles from 1969 to 2002.	Preserving natural dentition is essential to maintaining proper oral function. After dental extraction, we lose periodontal receptors that perceive important information during our oral function.
Abarca et al. [10] 2006	Studying the neurophysiological aspects of osseoperception.	Review of 93 articles from 1978 to 2003.	There is a tactile sensibility associated with dental implants that enables a physiological integration of the prosthesis and a more natural function of the rehabilitation.
Van Steenberghe D., Jacobs R. [11] 2006	Studying the influence of oral muscles in implant-supported prosthesis and its relationship with osseoperception.	Review of 28 articles from 1979 to 2006.	Muscular function in implant prosthesis was acceptable but presented lower forces during maximum function and a greater fatigue than in natural dentition.
Jacobs R, Van Steenberghe D. [3] 2006	Studying clinical integration of dental implants thanks to osseoperception.	Review of 57 articles from 1967 to 2005.	Osseoperception is responsible for the good integration and functioning of dental implant rehabilitations.
Klineberg et al. [12] 2005	Defining what osseoperception is and what kind of mechanoreceptors are important in this phenomenon.	A consensus statement about osseoperception.	Osseoperception may be defined as the sensation arising from mechanical stimulation of a bone-anchored prosthesis transduced by mechanoreceptors, together with a change in central neural processing in maintaining sensorimotor function.

**Table 2 medicina-58-00092-t002:** Clinical investigation results.

Author and Year	Objectives	Important Study Characteristics	Results
Shala KS. et al. [13] 2017	Measuring the threshold of tactile sensibility in patients wearing complete dentures.	88 patients wearing complete dentures participated in this study by biting thin metal foils.	The threshold of interocclusal perception in patients wearing complete dentures was higher than in natural dentition, and it kept decreasing thanks to the adaptation of the prosthesis.
Tanaka M. et al. [14] 2017	Measuring the masticatory adaptation after the rehabilitation with implants using immediate loading.	8 patients wearing implant prosthesis participated by biting pressure sensitive sheets	There was a gradual improvement in the biting force in patients wearing implant rehabilitations. There was no improvement in masticatory efficiency or in the perception of food hardness.
Bakshi P.V. et al. [15] 2017	Studying active tactile sensibility in patients wearing implant prosthesis and its evolution after prosthetic loading, then comparing these results with those from natural teeth.	20 subjects with different prosthetic rehabilitations had to perceive the absence or presence of articulating papers of varied thickness placed interocclusally.	There was a progressive improvement in tactile sensibility when wearing implant rehabilitations, and sensibility thresholds were very similar to those in natural dentition when their antagonist teeth were natural teeth.
Corpas Ldos et al. [16] 2014	Establishing the presence of nerve fibers surrounding dental implants.	Study of 12 failed implants that were removed from 10 patients. Then, a histological analysis of peri-implant bone was performed.	There was innervation around dental implants, and it was related to osseoperception, although its functioning and origin were not well known.
Reveredo A. et al. [17] 2013	Studying the active tactile sensibility in single dental implants by psychophysical tests.	20 subjects with implants and natural antagonistic teeth had to perceive thin foils placed interocclusally.	Implant prosthesis may resemble natural teeth in functioning thanks to osseoperception, which is the main advantage with respect to conventional dentures.
Kazemi et al. [18] 2013	Comparing active tactile sensibility values in dental implants and natural dentition.	25 subjects with implants and natural antagonistic and contralateral side teeth had to perceive different thickness foils.	Dental implants were slightly less sensitive to tactile stimulus than natural teeth.
Enkling et al. [19] 2012	Measuring tactile sensibility in single implants when their antagonists are natural teeth and are under anesthesia; later comparing the results with those obtained when measuring sensibility in natural dentition with one antagonist tooth anesthetized.	62 subjects were asked to bite on narrow copper foil varying in thickness and to decide whether or not they were able to identify a foreign body between their teeth.	Implants presented a similar sensibility with respect to natural dentition when their antagonists were under anesthesia, which entailed that implants presented an individual sensibility.
Habre-Hallage et al. [20] 2012	Studying the influences of osseoperception in brain cortex by using fMRI *.	9 patients with natural teeth and central incisor implants participated in this study. Teeth and implants were stimulated with a device connected to fMRI.	There was a cortical reprogramming after losing a tooth and replacing it with and implant that allowed a better functional integration of implant-supported prosthesis.
Enkling N. et al. [21] 2010	Describing active tactile sensibility in single implants with different surfaces.	62 subjects with single tooth implants and natural antagonistic teeth had to perceive thin copper foils placed interocclusally.	Active tactile sensibility in implants presented a low threshold very close to that present in natural teeth, and there were differences in values of sensibility between different implant surfaces.
Enkling N et al. [22] 2010	Studying active tactile sensibility in natural teeth.	68 complete dentulous subjects were asked to bite on thin copper foils of different thicknesses placed interocclusally.	Active tactile sensibility in natural teeth presented really low thresholds, so that tiny occlusal changes might have been perceived by patients, emphasizing the importance of a good occlusal adjustment in our rehabilitations.
Grieznis L. et al. [23] 2010	Comparing passive tactile sensibility between dental implants and natural teeth.	29 patients participated in this study. A pressure-sensitive device applied forces to implants and teeth.	Passive tactile sensibility in implants was lower than that present in natural dentition.
Batista M, Bonachela W, Soares J. [24] 2008	Comparing active tactile sensibility between dental implants and natural teeth.	70 subjects with different prosthetic rehabilitations were asked to bite aluminum foils with different thicknesses.	Complete dentures presented lower tactile sensibility than implant-supported prosthesis, the results of which were very similar to natural dentition.
Enkling et al. [25] 2007	Comparing active tactile sensibility between dental implants and natural teeth.	62 subjects with single tooth implants and natural antagonistic teeth had to bite thin copper foils placed interocclusally.	There were no significant differences in active tactile sensibility between natural dentition and dental implants when these presented a natural tooth as an antagonist.
El-Sheik A. et al. [26] 2004	Measuring passive tactile sensibility in implant prosthesis and relating the results with factors such as age, gender, or implant characteristics	20 subjects treated with mandibular implants were studied. A custom-made device applied pushing forces to implants until patients perceived pressure sensation.	Passive tactile sensibility values varied between different patients but they could not be related to the factors studied.
Hoshino K. et al. [27] 2004	Studying periodontal receptors response against dental implant as antagonist.	3 subjects with implant prosthesis participated in this study. A measuring device applied pulsations to implants.	Periodontal receptors from antagonistic teeth were not affected by implants, not even in the case of occlusal overload

* fMRI: functional magnetic resonance imaging.

## Data Availability

Not applicable.

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
