# Peer review of "Importance of Osseoperception and Tactile Sensibility during Masticatory Function in Different Prosthetic Rehabilitations: A Review"

_medicina, 2022, doi:10.3390/medicina58010092_

Round 1

Reviewer 1 Report

The idea of producing a magazine on this theme is very interesting, and the discussion is also well done. However, the methodology is poorly described and/or not sufficiently explained. Was there a double reading during the selection of the articles? Which selection procedure was chosen? Prisma? Is there a bottom-up reading? In table 1, in the type of studies, there appear rewiews, it is not a proper study, it should be separated the clinical studies and the rewiews. This work is interesting, but the restitution of the bibliographic search should be reviewed. 

Author Response

R1. Reply to comments. Modifications have been marked in the manuscript.

First of all, thank you for your comments, they have been a great help to improve our work.

We have changed the metodology of the study in order to describe and explain the review in a better way. Besides this, we have divided the table in 2 parts to achieve a more significant study. At first, we have created a table consisting of 10 review articles; and then we have made another one of 15 clinical research articles. We believe this organisation of articles may be more suitable for the requirements of this investigation.

We performed study selection according to the PRISMA (Preferred Reporting Items for Systematic Review and Meta-Analyses) guidelines for reporting systematic reviews.

The search strategy was conducted using the population, intervention, comparison, and outcome (PICO) framework based on the following question: “Do prostheses on implants have greater osseoperception than conventional prostheses??”

To answer this question, a sample population group of patients undergoing treatment with implant prostheses was selected. Controls were patients who were treated with conventional full dentures.

Regarding the selection process carried out, it was opted for the double reading of the articles by the research team.

Reviewer 2 Report

Dear Authors,

This a very well complied paper.

Please do look into these changes:

  1. Implantprosthesis is two word and maybe put as implant prosthesis.
  2. Line 121: Correct the syntax in the sentence.
  3. Line 222:  Correct the word Thea.
  4. Syntax in the table need to be checked.
  5. Results can be presented more elaborately.

Author Response

R2. Reply to comments. Modifications have been marked in the manuscript.

First of all, thank you for your comments, they have been a great help to improve our work.

We have improved the syntax of this article and we have corrected language failures.

We followed the advice of the assistant editor and got our article checked by a native English-speaker colleague.

Also, we presented our results more in detail; the results are now divided in 2 tables, one regarding reviews and another regarding clinical research.

We have added another column as a summary to describe every study more appropriately.

Reviewer 3 Report

General comments

It is a good review topic and it adds updated information for readers however, it needs improvement in English, use of scientific language and the focus of study needs to be succinate with each point verified regarding the selected studies. It is understood that tactile sensibility and osseoperception is important but why, studies needs to be explained more thoroughly with justifications given, and what is the significance of both terminologies or does it lead to any harms, this all should be stated straightaway in abstract and in introduction, basically the article needs a story line to follow.

Abstract

Gaps between words in abstract and throughout (such as implantprosthesis)

Which medical data bases, better to specify and extend the methodology on what basis did the authors searched the articles.

Abstract needs to be succinate, concise and with essential information that gives the summary of whole article, whereas, the abstract is too short at the moment and it needs more explanation to make it readable to audience in terms of understanding the concept of article.

Introduction

Page 2 line 20-21 after the finishing the sentence at system… its needs to follow a story line, that is if authors are talking about periodontium, talk about correlation between tactile sensibility and what causes the tactile sensibility and then follow periodontium sentence.

Sentence case used word ‘Century’ please use lower case- page 1 line 26, follow through out the manuscript

Page 1, line 31, thanks-use scientific language and define what is osseoperception,

Page 1word ‘prevalent’-use scientific language line 43

Page 2- line 52-59, add more references and add in more background to it

Material and methods

Was there a particular reason to add in studies from 2004 to 2017? Why not any recent ones?

Page 2 line 71- research studies- be specific what type of research studies

Spell check page 2 line 71

Explain the reason briefly in inclusion criteria as well regarding the chosen studies

Keywords and mesh terms, should be increased page 2 line 90-92

Computer resources- no need to mention the model of PC, etc.

Results

Only 2 studies? So at the end why 25?

In table, type of study can’t be stated as investigations in humans- it can RCT, original etc so please correct, the table needs more information on clinical studies that what was the population and how they came to results what are mentioned so I suggest please add more columns to make the table significant

Page 7 line 110- Corpas et al, please correct

Discussion

Page 7 line 128, page 8 line 156, 158,196,203- spell check, proof read is required throughout the manuscript.

Use of more scientific language and English correction is required- page 7, line 126, 130-134

It is fine to discuss and compare the studies in discussion however, to prove each study point, it needs to be justified through its methods and then results, please improve the section.

Conclusion

Please improve conclusion, it does not significantly explain the results of whole study

Author Response

R3. Reply to comments. Modifications have been marked in the manuscript.

First of all, thank you for your comments, they have been a great help to improve our work.

We followed your suggestions and got our manuscript checked by a native English-speaker colleague in order to improve the syntax and language failures.

We have elongated the abstract to make it more readable and understandable, specifying methodology and describing key points, as well as in the introduction.

In material and methods, we have detailed the types of research studies and inclusion/exclusion criteria. The search was carried out until 2021, but there is no recent article, the most innovative being the article from 2017.

In results, we talked about 2 review articles that were really useful for us to understand osseoperception but we cut off that part as it may lead to misunderstandings. Since a better explanation of the results was needed, we added another table and column to obtain an improved organisation of the studies. We divided the table in 2 parts, one referred to review articles and another referred to clinical research. The column summary helps to describe every investigation more clearly.

We made changes in the discussion too, in order to justify the results; although the lack of homogeneity of each article is a challenge we tried to solve with our changes in the organisation of results.

Finally, we modified the conclusion to improve the significance of the study. 

Round 2

Reviewer 1 Report

Thanks for the corrections, the manuscript is much improved and still interesting

Author Response

Thank you very much for your comments, they have been a great help to improve our work.

Reviewer 3 Report

Thank you for the  the revised version of manuscript by authors.    My final emphasis would be, to still improve the conclusion section and table 1 and 2 as the section summary in the table does not depict the summary of study rather it can be named as important study characteristics etc. 

Author Response

R2. Reply to comments. Modifications have been marked in the manuscript.

First of all, thank you for your comments, they have been a great help to improve our work.

At first, we have modified the column summary in both tables of results section as the previous denomination was not suitable. Important study characteristics defines in a better way the meaning of this column.

Finally, we have changed the conclusion section too, in order to explain in a better way the keypoints of this investigation and to support the results more appropriately.
